# OpenReview forum: "Chasing Moving Targets with Online Self-Play Reinforcement Learning for Safer Language Models"
_ICLR.cc/2026/Conference — Submitted to ICLR 2026_

### Official Review · Reviewer_mcqS · 2025-10-27

**Soundness:** 3
**Presentation:** 3
**Contribution:** 2
**Rating:** 4
**Confidence:** 3

**Summary:**

This paper introduces a novel online multi-agent reinforcement learning framework for improving the safety of Large Language Models (LLMs). The core idea is to move away from the conventional static, reactive patching of vulnerabilities towards a proactive, co-evolutionary process. The method frames LLM safety as a two-player, zero-sum game where a single, shared-parameter LLM alternates between an attacker role (generating adversarial prompts) and a defender role (safeguarding against them).

**Strengths:**

- It formulates red-teaming as a two-player zero-sum game with a formal safety guarantee at Nash Equilibrium.

- It shows strong empirical results, showing consistent gains across 12 benchmark and multiple model families and sizes.

- Extensive ablations show the effectiveness  of each proposed components.

**Weaknesses:**

- Reward model  and policy (defender and attacker) share the same parameter $\theta$, which looks confusing. Given that the WildGuard is used for reward model, it must be a notational mistake.  I think it would be better to use different parameters and explicitly state the reward model is frozen during entire training.

- The KL term in Eq. is undefined. I guess the authors might use token-wise reverse KL, but it would better to explicitly define the term for clarity.

- There is no direct head-to-head against red-teaming baselines. Rainbow Teaming [1] looks a relevant baseline if we use the same seed prompts for Rainbow Teaming.

- It is unclear why using the same backbone for both defender and attacker is helpful other than computational efficiency. The same model competes with conflicting objective. I wonder how the proposed method enables stable training and performance improvement.

- It heavily relies on initial seed prompts. Even though the attacker generated diverse attack prompts, they are still variants of the initial seed prompts, rather than new type of attacks.



## References

[1] Samvelyan, Mikayel, et al. "Rainbow teaming: Open-ended generation of diverse adversarial prompts." Advances in Neural Information Processing Systems 37 (2024): 69747-69786.

**Questions:**

- If we use a model that is already capable of generating think part, how the proposed method would work?

- What happens if seed prompts are not available for training?

- Are there any examples of reward hacking? The reward model is not perfect, which might lead to false harmful attack prompts.

---

> ### Author Response · Authors · 2025-11-26
> **Rebuttal (1/4)**
>
> Thank you for your positive and constructive feedback, especially for highlighting our *“extensive ablations”* and *“strong empirical results”*. We address your questions and comments in the following response, and we are happy to follow up on any point for further clarification during the discussion period.
>
> ---
>
> ## Clarification Around Shared Parameters
>
> Thanks for raising this critical point! You are absolutely right that the reward model is frozen during training, and that the Attacker and Defender agents share the parameters of the policy model. We realized that the confusion arose because the paper used $\theta$ in two different places: once to denote the reward model (in Section 3) and once to denote the shared attacker/defender policy model (in Section 4.2). **To remove this ambiguity, we have updated the notation in Section 3 to use $\phi$ for the reward model.**
>
> ---
>
> ## Clarification Around the KL Term
>
> Thanks for raising this point! We have indeed used the token-wise reverse KL divergence. We have replaced the notation and write it out explicitly with log-probabilities in Eqn. 1 in the revised paper.
>
> ---
>
> ## Clarification on Why We Co-Evolve Attacker and Defender Agents
>
> Thank you for raising this question and for recognizing that our method achieves computational efficiency by having the Attacker and Defender share the same policy model. We would like to offer the following perspectives to clarify why this design choice is an intentional and well-motivated feature, rather than a limitation.
>
> > ### Design Motivation and the Significance of Computation Efficiency
>
> First, our core goal in this project is to develop **an alternative end-to-end safety alignment method** that improves upon standard safety training approaches, which typically evolve a defender against a fixed, static set of attack prompts. Because of this, **we want our method to avoid introducing significant additional computational overhead beyond the standard setup.**
> Using two separate models would either double the memory requirement or require substantial on-and-off loading, both of which would complicate the pipeline and make training prohibitively expensive. Reducing the computational requirement is therefore a meaningful contribution, as it ensures the practicality and accessibility of a method that would otherwise be infeasible to run. In addition, we do not have the resources to support online training for two models simultaneously.
> For these reasons, we design our approach so that both attacker and defender roles are instantiated within a single model, retaining the benefits of coevolution while keeping the computational cost comparable to standard single-model safety training.
>
> > ### Role-specific Advantage Normalization is crucial for training stability
>
> We would like to clarify that, in practice, the trained models **reliably develop distinct attacker and defender personas** over the course of training. These roles are **explicitly separated through system and instruction prompts** that specify the intended behavior of each agent. Although the two agents share the same policy model, they operate in different modes that are determined by the system prompt. In addition, because we normalize advantages separately for the attacker and the defender during training, the model learns context dependent optimization: it learns how to attack when prompted as an attacker and how to defend when prompted as a defender, without experiencing catastrophic forgetting. As a result, the two agents do not end up competing with one another or pursuing conflicting objectives.
>
> > ### Motivation for Autonomous Self-Improvement
>
> In addition, we aim to investigate the self-improving potential of models with minimal reliance on accessory modules (except the reward model that’s standard to all RLHF algorithms). To fulfill this goal, we restrict the self-play agents to begin from the same initial state so the model can surface and correct its own blind spots, rather than overfitting to the attack distribution of an external adversary. This setup directly supports the “self-evolving paradigm” that we aimed to investigate in the paper, reflecting our goal of modeling how a system can iteratively refine its own weaknesses through autonomous coevolution.

---

> ### Author Response · Authors · 2025-11-26
> **Rebuttal (2/4)**
>
> ## Clarification Around Direct Comparison to Red-Teaming Methods
>
> Thanks for the suggestion to compare against red-teaming methods such as RainbowTeaming. We agree that RainbowTeaming is a powerful approach for generating diverse adversarial prompts. However, **it is important to distinguish between generating state-of-the-art attacks and achieving state-of-the-art safety alignment.** Our method is designed as an end-to-end safety training framework whose primary objective is to improve a model’s robustness. In this context, the co-evolving attacker acts as an auxiliary mechanism whose role is to expose weaknesses that help the defender improve, not to function as the strongest possible attacker. **Safety training is ultimately about the adaptability of the defender, not about maximizing attacker strength.**
>
> **Static vs. Dynamic Alignment.** Methods like RainbowTeaming operate as “offline” data generators: they produce a static adversarial dataset that is then used to fine-tune a model. Our **Defender-Only** baseline (Table 2), RL fine-tuned on the high-quality WildJailbreak attack set, mirrors this paradigm exactly. Despite using strong adversarial data, the Defender-Only model yields lower robustness gains (+30.26% average safety improvement over the baseline) than Self-Play (+43.68%).
>
> **The “Moving Target” Problem.** As discussed in our introduction, static training encourages overfitting to particular attack patterns, leading to “reactive patching.” In contrast, our co-evolution framework continually updates the attacker to exploit the defender’s current vulnerabilities. This dynamic curriculum prevents memorization and drives the defender toward more generalizable robustness.
>
> In summary, while RainbowTeaming is an excellent tool for generating adversarial attacks (**though its code is not open-sourced**), our objective is fundamentally different. Self-RedTeam is designed to train a more adaptive and resilient defender, and its online co-evolution mechanism provides a more effective pathway toward robust and generalizable safety alignment.
>
> ---
>
> ## Clarification Around the Reliance of the Seed Prompts
>
> We acknowledge the reviewer’s comment regarding seed prompts; however, we wish to clarify that **utilizing an initial state is a standard requirement across RL optimization methods (including RLHF and adversarial RL)**. In our framework, the seed prompts serve solely to define the harmful intent (e.g., “make a bomb”), leaving the attacker to explore the vast space of attack vectors surrounding that intent.
>
> Crucially, our empirical results demonstrate that the attacker does not simply regurgitate or paraphrase these seeds. As visualized in **Figure 2**, the Attacker Only baseline collapses into narrow SBERT clusters, whereas our Self-Play method induces a significantly broader and more diverse distribution. Furthermore, **Appendix D.4** reports a **+16.12% increase in perplexity** when evaluating the base model on attacker-generated queries. This statistical distinctiveness confirms that the generated attacks are novel distributions rather than simple rephrasings. Ultimately, while seeds provide the starting coordinates, the self-play dynamic drives the discovery of complex delivery mechanisms far beyond the initial inputs. This is evidenced by the improved robustness on downstream benchmarks, which confirms the defender was exposed to genuinely novel and challenging adversarial distributions during training.

---

> ### Author Response · Authors · 2025-11-26
> **Rebuttal (3/4)**
>
> ## “Use a model that is already capable of generating think part”
>
> We agree that evaluating our method on the latest generation of explicit “o1-style” reasoning models would be ideal. While we are actively working on these experiments, we could not include them in this rebuttal due to the short timeframe and significant technical complexities—specifically regarding vLLM version compatibility with our development system, strict thinking budget enforcement, and Out-Of-Memory (OOM) issues caused by extremely long context generations. However, **we believe our existing experiments with Qwen2.5 already serve as a strong proxy for answering this question.** Many recent works succeed in reinforcing the reasoning capability of Qwen2.5 to achieve spectacular performance on reasoning benchmarks [1,2,3]. Additionally, we infer that **Qwen2.5 is effectively a "reasoning-capable" model** based on the following evidence:
> 1. **Empirical "Thinking" Behavior:** Our training dynamics suggest the base Qwen2.5 shares this reasoning foundation. When applying our method, Qwen2.5 adapted almost immediately, starting with a low **~10% format CoT template violation rate** and converging to near-zero within just 10 iterations. This stands in stark contrast to Llama 3.1, which behaved like a model "learning to think" from scratch (starting with **~80% CoT template violations** and requiring significantly more steps).
> 2. **Connection to Qwen2.5-Math:** The math-specialized variant, `Qwen2.5-Math`, is confirmed to be post-trained with CoT reasoning traces and supports the CoT format out-of-the-box.
>
> ​​These results demonstrate that for models already capable of generating thought (like Qwen2.5), our method functions as an **alignment mechanism**. Rather than synthesizing reasoning from scratch, it efficiently steers the model's pre-existing reasoning capabilities into the target format required by the reward function.
>
> [1] *DeepSeek-R1: Incentivizing Reasoning Capability in LLMs via Reinforcement Learning*, https://arxiv.org/pdf/2501.12948
>
> [2] *Reinforcement Learning for Reasoning in Large Language Models with One Training Example*, https://arxiv.org/pdf/2504.20571
>
> [3] *Does Reinforcement Learning Really Incentivize Reasoning Capacity in LLMs Beyond the Base Model?*, https://arxiv.org/pdf/2504.13837

---

> ### Author Response · Authors · 2025-11-26
> **Rebuttal (4/4)**
>
> ## Discussion of Potential Reward Hacking Behaviors
> You are absolutely right that reward hacking is a well known challenge in RL, and if the judge or reward model contains blind spots, RL optimization may exploit them. To address this concern, we investigate the issue from several complementary angles and summarize these analyses and mitigation strategies below.
>
> > ### New results with Qwen3Guard as the reward model
>
> The Self-RedTeam framework is not tied to any particular reward model. Its modular structure supports plug-and-play integration of alternative and potentially stronger reward models as they emerge. We chose WildGuard for our main experiments because it was the only model that provided rating modes for both harmfulness and refusals, which allowed us to avoid loading two separate models. To demonstrate the flexibility of Self-RedTeam, we also trained models using Qwen3Guard `(Qwen/Qwen3Guard-Gen-8B)` as the reward model and found conclusions that aligned with those obtained using WildGuard. This provides direct evidence to show that our method demonstrates generalizable improvements across different reward models.
>
> Results below are averaged over three runs to verify robustness. Replacing WildGuard with Qwen3Guard consistently improves performance, which is expected given that Qwen3Guard is a significantly stronger, recently released (last month) model.
>
> | **Model** | **WG:test AH ↓** | **WG:test VH ↓** | **WJB AH ↓** | **DAN ↓** | **HarmBench AH ↓** | **HarmBench VH ↓** | **OR-Bench VH ↑** | **XSTest VH ↑** | **StrongREJECT VH ↑** | **WJB AB ↑** | **XSTest VB ↑** | **Alpaca-Eval 2 ↑** |
> |---|---|---|---|---|---|---|---|---|---|---|---|---|
> | Qwen2.5-7B-Instruct | 0.303 | 0.027 | 0.864 | 0.390 | 0.278 | 0.163 | 0.879 | 0.890 | 0.920 | **0.992** | 0.948 | 33.428 |
> | Self-RedTeam - WildGuard | 0.179 | 0.002 | 0.489 | 0.222 | 0.161 | **0.044** | 0.968 | 0.912 | 0.979 | 0.979 | **0.960** | **34.927** |
> | Self-RedTeam - **Qwen3Guard** | **0.156** | **0.000** | **0.441** | **0.211** | **0.142** | **0.044** | **0.978** | **0.922** | **0.983** | 0.976 | 0.949 | 33.835 |
>
> > ### Self-play coevolution is inherently more robust to reward hacking compared to evolving a single agent
>
> Self-play naturally mitigates the model from collapsing into a single gamed mode because the target is continually shifting. As soon as the attacker exploits a weakness, the defender adapts, which pushes the attacker to move in new directions rather than overfitting to one specific weakness in the reward model.
>
> We can observe these training dynamics in Figure 3 of the paper. The Attacker Only baseline shows declining revision faithfulness (around 60 percent in Figure 3d) despite an increasing harmfulness win rate (Figure 3b), because it begins to optimize for higher win rates instead of following instructions. In contrast, the Self-play model maintains high attack revision faithfulness throughout training, which highlights its stronger robustness against reward hacking. Also as shown in Figure 1, the Attacker-Only baseline suffers from mode collapse, converging to a dominant strategy (e.g., 'misinformation campaigns') regardless of the context, which confirms the presence of reward hacking. Conversely, our Self-Play method prevents this collapse; the continuous adaptation of the opponent creates a dynamic environment that precludes the attacker from exploiting a static strategy.

---

### Official Review · Reviewer_4Pif · 2025-10-31

**Soundness:** 3
**Presentation:** 3
**Contribution:** 2
**Rating:** 6
**Confidence:** 2

**Summary:**

This paper proposes SELF-REDTEAM, an online self-play reinforcement learning framework for LLM safety alignment, where a single model alternates between attacker and defender roles with a hidden Chain-of-Thought (CoT). The idea of framing safety training as a zero-sum game between co-evolving agents is novel and theoretically grounded, providing a clear motivation for proactive rather than reactive safety alignment.

**Strengths:**

Strengths:

- Conceptually appealing formulation of LLM safety as a self-play MARL problem with a Nash equilibrium–based safety guarantee.

- Solid empirical results across multiple model families (Llama, Qwen), demonstrating significant robustness gains (up to 95% ASR reduction) with minimal performance degradation.

- The Hidden CoT mechanism is an elegant addition, improving attack diversity and mitigating over-refusal.

**Weaknesses:**

Weaknesses:

- The theoretical guarantee relies heavily on the quality of the reward model; practical convergence to Nash equilibrium is not verified.

- Some evaluation benchmarks (e.g., WildGuard/WildJailBreak) overlap with training data, potentially inflating results.

- Experimental section could be more transparent about compute cost and stability during training.

**Questions:**

1. How do you measure or verify convergence toward the proposed Nash Equilibrium in practice?

2. How sensitive are the results to the choice or bias of the reward model used for safety evaluation?

3. Does the observed 95% ASR reduction generalize to unseen or multi-turn adversarial prompts?

---

> ### Author Response · Authors · 2025-11-26
> **Rebuttal (1/3)**
>
> Thank you for your positive and constructive feedback, especially for highlighting that our method formulation is *“conceptually appealing”*, our empirical results are solid, and the hidden CoT design is elegant. We address your questions and comments in the following response, and we are happy to follow up on any point for further clarification during the discussion period.
>
> ---
> ## Discussion of the Empirical Convergence regarding the Theoretical Nash Equilibrium (NE)
>
> Thank you for raising this point. We agree that our NE discussion can benefit from further clarification. Our intention was to invoke NE to formalize the training objective of our co-evolutionary setup, rather than to claim provable convergence to an exact equilibrium. Below, we expand on both the theoretical motivation and the empirical evidence regarding convergence.
>
> > ### NE Serves as the Theoretical Objective, Not a Guaranteed End State
>
> As established in Theorem 1, optimizing a single agent against a fixed opponent, e.g., training a defender against a static attack set or training an attacker against a frozen-weight defender, fails to provide the safety guarantees inherent to an equilibrium. This motivates our use of self-play. While self-play allows us to explore the attack-defense surface more effectively than static training, **guaranteeing convergence to a global Nash Equilibrium remains a fundamental challenge in non-convex deep learning optimization like LLM fine-tuning.** Thus, the NE analysis serves as a theoretical "North Star" guiding the training trajectory, rather than a guaranteed end state.
>
> > ### Empirical Results Validate the Theory
>
> Despite the optimization challenges, we observe strong empirical stability. As shown in Figure 3, the training dynamics fluctuate and then gradually settle into a fixed point where the defender’s win rate approaches $100\%$. **This empirical outcome mirrors the properties of the theoretical NE**, where an optimal defender is expected to robustly counter the attacker.
>
> > ### Robust Safety Improvements Without Full Empirical Convergence
>
> To empirically examine this, we conducted exhaustive evaluations across a broad spectrum of adversarial settings, **including all 12 static benchmarks, newly generated dynamic attacks, and dynamic multi-turn adversaries such as X Teaming**. Together, these evaluations stress test the model far beyond its training distribution and provide an empirical approximation of the diverse adversarial space envisioned in the theoretical equilibrium.
>
> Across all these benchmarks, **we observe consistent and substantial safety improvements**, even though the training dynamics do not perfectly converge to a Nash Equilibrium. This shows that exact convergence is not necessary for the method to yield meaningful and robust gains. In practice, the model reliably moves toward the desirable equilibrium region of the strategy space, achieving strong generalization and robustness despite the theoretical difficulty of reaching a formal NE.

---

> ### Author Response · Authors · 2025-11-26
> **Rebuttal (2/3)**
>
> ## Self-RedTeam is Generalizable Across Different Reward Models (New Results with Qwen3Guard as the reward model)
>
> You are right that the empirical convergence may rely on the quality of the reward model. The Self-RedTeam framework is not tied to any particular reward model. Its modular structure supports plug-and-play integration of alternative and potentially stronger reward models as they emerge. We chose WildGuard for our main experiments because it was the only model that provided rating modes for both harmfulness and refusals, which allowed us to avoid loading two separate models. To demonstrate the flexibility of Self-RedTeam, we also trained models using Qwen3Guard `(Qwen/Qwen3Guard-Gen-8B)` as the reward model and **found conclusions that aligned with those obtained using WildGuard.** This provides direct evidence to show that our method demonstrates generalizable improvements across different reward models.
>
> Results below are averaged over three runs to verify robustness. **Replacing WildGuard with Qwen3Guard consistently improves performance**, which is expected given that Qwen3Guard is a significantly stronger, recently released (last month) model.
>
> | **Model** | **WG:test AH ↓** | **WG:test VH ↓** | **WJB AH ↓** | **DAN ↓** | **HarmBench AH ↓** | **HarmBench VH ↓** | **OR-Bench VH ↑** | **XSTest VH ↑** | **StrongREJECT VH ↑** | **WJB AB ↑** | **XSTest VB ↑** | **Alpaca-Eval 2 ↑** |
> |---|---|---|---|---|---|---|---|---|---|---|---|---|
> | Qwen2.5-7B-Instruct | 0.303 | 0.027 | 0.864 | 0.390 | 0.278 | 0.163 | 0.879 | 0.890 | 0.920 | **0.992** | 0.948 | 33.428 |
> | Self-RedTeam-WildGuard | 0.179 | 0.002 | 0.489 | 0.222 | 0.161 | **0.044** | 0.968 | 0.912 | 0.979 | 0.979 | **0.960** | **34.927** |
> | Self-RedTeam-**Qwen3Guard** | **0.156** | **0.000** | **0.441** | **0.211** | **0.142** | **0.044** | **0.978** | **0.922** | **0.983** | 0.976 | 0.949 | 33.835 |
>
> ---
>
> ## Clarification of Evaluation Benchmark Overlaps and Results Generalizability
>
> Thank you for raising the concern regarding potential inflation of evaluation results due to overlap between training and evaluation data. **We want to clarify that there is no overlap between the data used for training and the data used for evaluation.** Our training pipeline utilizes "vanilla" harmful seeds (e.g., simple prompts such as “how to create a bomb?”) strictly drawn from the WildJailbreak **training** split. The attacker is then tasked to transform these seeds into adversarial attacks, which are substantially more complex but preserve the same harmful intent. Crucially, our evaluation is performed solely on the "adversarial" harmful prompts from WildJailbreak **test** split. This has ensured that the model never encounters the specific adversarial queries used for testing.
>
> To further demonstrate generalizability, **we evaluate across 12 diverse safety benchmarks that span a broad range of tasks.** In addition, **we apply dynamic attack methods** from X-Teaming and HarmBench (including PAIR, AutoDAN, and more to be added) to test robustness under adaptive, multi-step adversaries. The consistent robustness gains across these **out-of-distribution settings** confirm that our method effectively generalizes to unseen attacks and adaptive adversaries, and out-performing off-the-shelf models aligned with standard RLHF. We had added these new results to the Appendix C.4 of the revised paper.
>
> Below, we provide the full results for **X-Teaming, precomputed HarmBench (adversarial harmful)**, and the **new dynamic evaluations**. Similarly we observe significant ASR reduction up to 50.7% on PAIR and AutoDAN.
> | Models | X-Teaming | precomputed HarmBench (adversarial harmful) | PAIR | AutoDAN |
> |---|---|---|---|---|
> | Qwen2.5-3B-IT | 88.0 | 0.265 | 0.297 | 0.5175 |
> | +Self-RedTeam | 62.0 | 0.178 | 0.148 | 0.452 |
> | +Impv% | +29.5 | +33.0 | +50.0 | +12.7 |
> | Qwen2.5-7B-IT | 88.0 | 0.278 | 0.305 | 0.555 |
> | +Self-RedTeam | 80.0 | 0.161 | 0.185 | 0.3358 |
> | +Impv% | +9.09 | +42.1 | +39.3 | +39.5 |
> | Qwen2.5-14B-IT | 76.0 | 0.131 | 0.218 | 0.17 |
> | +Self-RedTeam | 66.0 | 0.064 | 0.1075 | 0.105 |
> | +Impv% | +13.2 | +51.6 | +50.7 | +38.2 |

---

> ### Author Response · Authors · 2025-11-26
> **Rebuttal (3/3)**
>
> ## Clarification of Computation Cost and Stability
>
> We appreciate the suggestion to elaborate on these aspects. Regarding computational cost, we have provided a detailed analysis in **Appendix D.3** and disclosed the full training setup (including runtime and GPU specifications) in **Appendix A.2**.
>
> Regarding training stability, we address this from both methodological and empirical perspectives:
> 1. **Methodological Stability:** As noted in Section 4.2 ("RL training objectives"), the use of normalized role-specific token-level advantages is critical to our framework. This intuitive mechanism allows a single model to learn effectively from the conflicting reward signals of the attacker and defender roles simultaneously, preventing the optimization instability often seen in multi-agent setups.
> 2. **Empirical Convergence:** In Section 6, our analysis of learning dynamics shows that this mechanism works: training remains stable across runs, with agents co-evolving toward consistent behavioral modes without oscillatory collapse. To ensure statistical robustness, all results reported in the paper are averaged over at least three independent runs.
>
> **We have revised the manuscript to explicitly link the technical design in Section 4.2 to the stable dynamics observed in Section 6.**

---

### Official Review · Reviewer_h9EC · 2025-11-05

**Soundness:** 3
**Presentation:** 4
**Contribution:** 3
**Rating:** 6
**Confidence:** 4

**Summary:**

This paper introduces Self-RedTeam, an online self-play RL algorithm that trains a single model to co-evolve attacker and defender roles in a two-player zero-sum game, generating adversarial prompts and safeguarding against them. They show that if the models reach Nash Equilibrium, the model is theoretically guaranteed to be safe — although this is likely impossible to achieve in practice. The approach demonstrates empirical improvements across multiple safety benchmarks (including WildGuard and HarmBench), and two model families (Llama3, Qwen2.5). The authors also offer a nice study on the distribution of the attacks generated by their method, showcasing how their method generates more diverse attacks over only fine-tuning an attacker LLM against a static defender. I have a few comments with regards to the theoretical justification as well as the evaluations (which could potentially be improved), but overall I am leaning towards a weak accept as the approach is nice, the work is very polished and well executed, and would be of interest to the research community.

**Strengths:**

1. The method is practical and efficient with a thorough analysis on the overhead of their approach (~45% longer than baseline with online generation). The framework is general and can be applied to any safety training pipeline, with a reasonable improvement to the refusal rate on the models tested.
2. The paper is very well written and polished; the authors conduct many experimental results, including comparisons to other safeguarding baselines (LAT, CircuitBreakers), as well as ablations on the various components of their approach (self-play, CoT, SFT). The appendix is detailed and provides all necessary implementation details one would need to reproduce their work.

**Weaknesses:**

1. If I understood correctly, the evaluations were done against *static* adversarial prompts (with the exception of X-teaming); stronger non-static attacks should be considered for the evaluations (i.e. applying some of the algorithmic methods to the final trained model itself, rather than using the preexisting attacks on other models). If the paper is indeed missing these evals, I would strongly recommend them for the discussion period.
2. Results indicate that the improvements are decent but not spectacular; some evals have good improvements but others are very modest; I think it would be critical to see how well this approach fares to what was discussed in W1.
3. The observation that over time, the attacker sometimes refuses to generate harmful attacks does suggest that it might not be ideal to use the same attacker/defender model
4. I understand that the Nash Equilibrium arguments are just to show that you are optimizing for the correct target. However, it feels a bit ad-hoc because it was presented as a justification for the approach, then acknowledged that convergence is unlikely in practice, and then no longer discussed. I think it would be nice to have more discussion of either how close the training can converge to the NE in practice, and/or what happens when it doesn’t.

**Questions:**

1. Re: emergent attacker refusal, it does feel like the attack/defense are in tension because the model is rewarded for breaking itself whilst also trying to be come more robust. Intuitively I would expect the results to improve if you had different models for the attacker and defender; have you experimented with this and seen anything to verify/disprove this hypothesis? How does this tie in to your theoretical motivation, as you require need strong attacker for robust defense?
2. The reliance on WildGuard 7B to evaluate components used in the reward could have biases and be prone to reward hacking; it would be nice to see some discussion on how sensitive the method is to the judge model’s quality/biases, what happens if the judge has exploitable weaknesses, and if attacks are able to game the specific reward model.
3. Why does the CoT help more for Llama than Qwen models?
4. Do attacks transfer to other LLMs?

---

> ### Author Response · Authors · 2025-11-26
> **Rebuttal (1/5)**
>
> Thank you for your positive and constructive feedback, especially for recognizing our method being “practical and efficient”, the analysis and experiments being thorough, and the paper being well-written and polished. We address their questions and comments in the following response, and are happy to follow-up on any point for further clarification during the discussion period.
>
> -----
>
> ## Dynamic Evaluations and Generalizability of the Evaluations
>
> Thank you for suggesting dynamic evaluations. In our original submission, we did not run exhaustive dynamic evaluations due to their substantial computational cost and inherent stochasticity. Nevertheless, we made two efforts to approximate dynamic evaluation.
>
> First, we included results on **X-Teaming**, one of the widely used dynamic benchmarks, and showed that our method yields **consistent, generalizable gains** in multi-turn adversarial settings over off-the-shelf models (improvements ranging from 9.09% to 44.7%). We also really appreciated that you have acknowledged our effort of testing it.
>
> Second, we curated a dataset of 1,500 adversarial attacks using the HarmBench framework (https://github.com/centerforaisafety/HarmBench). This collection spans 15 dynamic attack algorithms, five model-agnostic and ten model-dependent, including AutoDAN, AutoPrompt, EnsembleGCG, FewShot, GBDA, GCG, PAIR, PEZ, TAP, UAT, DirectRequest, HumanJailbreaks, IntentMasking, PAP, and ZeroShot. To balance computational feasibility with broad coverage, the model-dependent attacks were sampled across 22 open-source models. We utilize these pre-generated attacks as a static adversarial evaluation set. These results are presented in Tables 1 and 2 under the “HarmBench (adversarial harmful)” column, with further implementation details provided in Appendix C.1.
>
> That said, we fully agree that comprehensive **dynamic** evaluations provide important evidence of the generalizability of our training approach. In response, we additionally evaluated our models with the dynamic multi-turn attack methods available in HarmBench framework (including PAIR, AutoDAN). These results demonstrate that **models trained with our method consistently achieve higher adversarial robustness than off-the-shelf models trained with standard RLHF alignment.**
>
> Below, we provide the full results for **X-Teaming, precomputed HarmBench (adversarial harmful)**, and the **new dynamic evaluations**. Similarly we observe significant ASR reduction up to 50.7% on PAIR and AutoDAN. We have added these results to Appendix C.4 and included pointers in the main paper directing readers to these discussions.
> | Models | X-Teaming | precomputed HarmBench (adversarial harmful) | PAIR (**NEW**) | AutoDAN (**NEW**) |
> |---|---|---|---|---|
> | Qwen2.5-3B-IT | 88.0 | 0.265 | 0.297 | 0.5175 |
> | +Self-RedTeam | 62.0 | 0.178 | 0.148 | 0.452 |
> | +Impv% | **+29.5** | **+33.0** | **+50.0** | **+12.7** |
> | Qwen2.5-7B-IT | 88.0 | 0.278 | 0.305 | 0.555 |
> | +Self-RedTeam | 80.0 | 0.161 | 0.185 | 0.3358 |
> | +Impv% | **+9.09** | **+42.1** | **+39.3** | **+39.5** |
> | Qwen2.5-14B-IT | 76.0 | 0.131 | 0.218 | 0.17 |
> | +Self-RedTeam | 66.0 | 0.064 | 0.1075 | 0.105 |
> | +Impv% | **+13.2** | **+51.6** | **+50.7** | **+38.2** |
>
> ---
>
> > ## Clarification Around “Some Results Are Not Spectacular”
>
> While the reviewer noted that some improvements appear modest, we would like to recontextualize the notion of “spectacular” by grounding it in the fundamental safety–utility trade-off. A key challenge in safety alignment is that many existing methods (such as LAT) substantially compromise a model’s general capabilities. In contrast, our method preserves utility while delivering meaningful robustness gains: for example, Table 12 shows that Self-RedTeam maintains an AlpacaEval score of 22.04, compared to LAT’s 9.68, demonstrating that our approach avoids the severe capability collapse seen in prior work. At the same time, the safety improvements are empirically significant. For Llama-3.1-8B, we reduce the ASR on WildGuard Adv Harm from 0.237 (base model) to 0.080, which corresponds to a **$\sim$66% reduction in successful attacks**. Our methods also work particularly well with Qwen2.5 models, with safety improvement up to 100% and increase in instruction following marked by 0.3%~4.5% percentage gain on AlpacaEval-2. This combination of preserving utility while achieving large relative safety gains demonstrates the practical strength of our method, even if raw absolute deltas on some benchmarks appear moderate in isolation. In addition, the newly added dynamic evaluations further shows the consistent and robust improvement trend by applying our method to training models.

---

> ### Author Response · Authors · 2025-11-26
> **Rebuttal (2/5)**
>
> ## Discussion of Potential Reward Hacking Behaviors
>
> You are absolutely right that reward hacking is a well known challenge in RL, and if the judge or reward model contains blind spots, RL optimization may exploit them. To address this concern, we investigate the issue from several complementary angles and summarize these analyses and mitigation strategies below.
>
> > ### New results with Qwen3Guard as the reward model
>
> The Self-RedTeam framework is not tied to any particular reward model. Its modular structure supports plug-and-play integration of alternative and potentially stronger reward models as they emerge. We chose WildGuard for our main experiments because it was the only model that provided rating modes for both harmfulness and refusals, which allowed us to avoid loading two separate models. To demonstrate the flexibility of Self-RedTeam, we also trained models using Qwen3Guard `(Qwen/Qwen3Guard-Gen-8B)` as the reward model and **found conclusions that aligned with those obtained using WildGuard.** This provides direct evidence to show that our method demonstrates generalizable improvements across different reward models.
>
> Results below are averaged over three runs to verify robustness. **Replacing WildGuard with Qwen3Guard consistently improves performance**, which is expected given that Qwen3Guard is a significantly stronger, recently released (last month) model.
>
> | **Model** | **WG:test AH ↓** | **WG:test VH ↓** | **WJB AH ↓** | **DAN ↓** | **HarmBench AH ↓** | **HarmBench VH ↓** | **OR-Bench VH ↑** | **XSTest VH ↑** | **StrongREJECT VH ↑** | **WJB AB ↑** | **XSTest VB ↑** | **Alpaca-Eval 2 ↑** |
> |---|---|---|---|---|---|---|---|---|---|---|---|---|
> | Qwen2.5-7B-Instruct | 0.303 | 0.027 | 0.864 | 0.390 | 0.278 | 0.163 | 0.879 | 0.890 | 0.920 | **0.992** | 0.948 | 33.428 |
> | Self-RedTeam-WildGuard | 0.179 | 0.002 | 0.489 | 0.222 | 0.161 | **0.044** | 0.968 | 0.912 | 0.979 | 0.979 | **0.960** | **34.927** |
> | Self-RedTeam-**Qwen3Guard** | **0.156** | **0.000** | **0.441** | **0.211** | **0.142** | **0.044** | **0.978** | **0.922** | **0.983** | 0.976 | 0.949 | 33.835 |
>
> > ### Self-play coevolution is inherently more robust to reward hacking compared to evolving a single agent
>
> Self-play naturally mitigates the model from collapsing into a single gamed mode because **the target is continually shifting**. As soon as the attacker exploits a weakness, the defender adapts, which pushes the attacker to move in new directions rather than overfitting to one specific weakness in the reward model.
>
> We can observe these training dynamics in Figure 3 of the paper. The Attacker Only baseline shows declining revision faithfulness (around 60 percent in Figure 3d) despite an increasing harmfulness win rate (Figure 3b), because it begins to optimize for higher win rates instead of following instructions. In contrast, the Self-play model maintains high attack revision faithfulness throughout training, which highlights its stronger robustness against reward hacking.
>
> > ### Counting overrefusal bias
>
> Finally, in our original algorithm design, we directly countered overrefusal bias, which is a common judge weakness, by including benign prompts (50% in the seed prompt mix) and explicit refusal accuracy related sub-rewards.
> Taken together, these observations suggest that Self-RedTeam generalizes well across different reward models, and that both the Self-play design and the fine grained reward components help mitigate reward hacking. Thanks to your suggestion and **accordingly we have updated the paper and added these discussions to the Appendix D.6.**

---

> ### Author Response · Authors · 2025-11-26
> **Rebuttal (3/5)**
>
> ## Clarification on Why We Co-Evolve Self-Play Agents Instead of Training Two Separate Agents
>
> Thank you for suggesting that we train separate attackers and defenders. This is indeed a valuable direction, and our theory in fact can generalize to training separate attackers and defenders. However, we consider this to be a future line of investigation for the following reasons, and we have added these discussions to the Appendix E in the revised paper for better motivating our method design.
>
> > ### Design Motivation and Accessibility Constraints
>
> First, our core goal in this project is to **develop an alternative end-to-end safety alignment method** that improves upon standard safety training approaches, which typically evolve a defender against a fixed, static set of attack prompts. Because of this, **we want our method to avoid introducing significant additional computational overhead beyond the standard setup.** Using two models would either double the memory requirement or force substantial on-and-offloading, which would complicate the pipeline and reduce accessibility. In practice, we also do not have the resources to run online training for two models simultaneously. Therefore, we design our approach so that both attacker and defender roles are instantiated within a single model, allowing us to retain the benefits of coevolution while keeping the computational cost comparable to standard single-model safety training.
>
> > ###  Motivation for Autonomous Self-Improvement
>
> In addition, we aim to investigate the self-improving potential of models with minimal reliance on accessory modules (except the reward model that’s standard to all RLHF algorithms). To fulfill this goal, we restrict the self-play agents to begin from the same initial state so the model can surface and correct its own blind spots, rather than overfitting to the attack distribution of an external adversary. This setup directly supports the **“self-evolving paradigm”** that we aimed to investigate in the paper, reflecting our goal of modeling how a system can iteratively refine its own weaknesses through autonomous coevolution.
>
> > ### Attacker Refusals are a Llama-Specific Artifact, Not a General Phenomenon
>
> We clarify that, in general, trained models reliably evolve into distinct attacker and defender personas as defined by our system prompts. The attacker-refusal behavior noted in the limitations is specific to the Llama family’s safety alignment and is not a universal phenomenon.
> Our quantitative analysis confirms that this issue is absent in other model families. We sampled 3,000–5,000 revised attacks from three checkpoints (training iterations 100–200) for both Llama3.1 and Qwen2.5 models, using Qwen3-30B-A3B-Thinking-2507 as the judge for refusal classification. As shown in the table below, **all Qwen models (3B/7B/14B) exhibited a negligible refusal rate (<0.11%)**. In contrast, Llama-3.1-8B-Instruct exhibited a refusal rate of 25.27%, confirming that high refusal rates are an idiosyncrasy of Llama rather than a failure of the training framework. We attribute this to the Llama3.1-8B-IT base policy’s known propensity for over-refusal compared to Qwen, which occasionally persists in the attacker policy during coevolution. We have explicitly clarified this behavior in the revised manuscript.
>
> | Model | Refusal Rate % (Refusals/Total Tested) |
> | :--- | :--- |
> | Llama3.1-8B-Instruct-AB | 7.21% (339/4705) |
> | Llama3.1-8B-Instruct | 25.27% (1144/4527) |
> | Qwen2.5-3B-Instruct | 0.10% (3/3121) |
> | Qwen2.5-7B-Instruct | 0.04% (2/4691) |
> | Qwen2.5-14B-Instruct | 0.11% (5/4698) |
> * Noted that these aren't base models, but Self-RedTeam checkpoints. For convenience, we simply abbreviated them using their base model names.
>
> > ### Attacker Refusals Do Not Undermine the Core Objective: Improving Defender Robustness
>
> Finally, stepping back, the primary goal of our method is to **strengthen the defender’s robustness**. The **co-evolving attacker functions as an auxiliary component whose purpose is to help the defender improve, rather than being a standalone objective.** Even though Llama3.1-8B-IT exhibits a small number of attacker refusals, we still observe consistent gains on downstream evaluations. These improvements demonstrate that the central goal of our method, i.e., enhancing defender robustness, remains fully achieved.

---

> ### Author Response · Authors · 2025-11-26
> **Rebuttal (4/5)**
>
> ## Discussion of the Empirical Convergence regarding the Theoretical Nash Equilibrium (NE)
> ---
> Thank you for raising this point. We agree that our NE discussion can benefit from further clarification. Our intention was to invoke NE to formalize the training objective of our co-evolutionary setup, rather than to claim provable convergence to an exact equilibrium. Below, we expand on both the theoretical motivation and the empirical evidence regarding convergence.
>
> > ### NE Serves as the Theoretical Objective, Not a Guaranteed End State
>
> As established in Theorem 1, optimizing a single agent against a fixed opponent, e.g., training a defender against a static attack set or training an attacker against a frozen-weight defender, fails to provide the safety guarantees inherent to an equilibrium. This motivates our use of self-play. While self-play allows us to explore the attack-defense surface more effectively than static training, **guaranteeing convergence to a global Nash Equilibrium remains a fundamental challenge in non-convex deep learning optimization like LLM fine-tuning.** Thus, the NE analysis serves as a theoretical "North Star" guiding the training trajectory, rather than a guaranteed end state.
>
> > ### Empirical Results Validate the Theory
>
> Despite the optimization challenges, we observe strong empirical stability. As shown in Figure 3, the training dynamics fluctuate and then gradually settle into a fixed point where the defender’s win rate approaches $100\%$. **This empirical outcome mirrors the properties of the theoretical NE**, where an optimal defender is expected to robustly counter the attacker.
>
> > ### Robust Safety Improvements Without Full Empirical Convergence
>
> To empirically examine this, we conducted exhaustive evaluations across a broad spectrum of adversarial settings, **including all 12 static benchmarks, newly generated dynamic attacks, and dynamic multi-turn adversaries such as X Teaming**. Together, these evaluations stress test the model far beyond its training distribution and provide an empirical approximation of the diverse adversarial space envisioned in the theoretical equilibrium.
>
> Across all these benchmarks, **we observe consistent and substantial safety improvements**, even though the training dynamics do not perfectly converge to a Nash Equilibrium. This shows that exact convergence is not necessary for the method to yield meaningful and robust gains. In practice, the model reliably moves toward the desirable equilibrium region of the strategy space, achieving strong generalization and robustness despite the theoretical difficulty of reaching a formal NE.

---

> ### Author Response · Authors · 2025-11-26
> **Rebuttal (5/5)**
>
> ## Why do CoT help Llama more than Qwen?
>
> We hypothesize that this performance differential stems from **differences in baseline reasoning and instruction-following capabilities.** Standard Llama3.1 models generally exhibit weaker instruction adherence compared to Qwen2.5. Consequently, the structural scaffolding provided by CoT yields a larger marginal benefit for Llama. In contrast, Qwen2.5 possesses stronger intrinsic reasoning capabilities, resulting in diminishing returns from hidden CoT. Although it is unclear whether Qwen2.5 have been fine-tuned with CoT reasoning traces, its math-focused variant Qwen2.5-Math supports CoT reasoning format out-of-the-box.
>
> Our training dynamics support this hypothesis: Llama struggles to adopt the intended reasoning format—starting with an **~85% R1 CoT template violation rate** and reducing to near zero slowly within 30 iterations. Conversely, Qwen adapts almost immediately, starting with **a ~10% violation rate** that drops to near-zero within 10 iterations.
>
> ---
> ## Transferability of Attacks Across Models
>
> We appreciate this suggestion. To examine attack transferability, we generated adversarial prompts using Llama-3.1-8B-IT to target Qwen-2.5-7B-IT, and vice-versa, and use their attacks against Mistral-7B-Instruct-v0.3 and against self. We evaluated a set of 120 prompts sampled across three training checkpoints for each model.
> As shown in the table below, our method demonstrates strong transferability. Attacks generated by Qwen achieved a **55.3% Attack Success Rate (ASR)** against Llama, and Llama-generated attacks achieved a **54.8% ASR** against Qwen. Notably, these attacks generalized even more effectively to a held-out model, **Mistral-7B-Instruct-v0.3**, achieving nearly **80% ASR** regardless of the source model. The attacks become significantly less effective against checkpoints that have been similarly self-redteaming fine-tuned.
>
> | Attacker (⚔️) ASR | ⚔️ vs Llama3.1-8B-IT | ⚔️ vs Qwen2.5-7B-IT | ⚔️ vs Mistral-7B-Instruct-v0.3 | ⚔️ vs itself @ 100 iter |
> | :--- | :---: | :---: | :---: | :---: |
> | **Self-RedTeam (Qwen2.5-7B-IT)** | 55.3% | 54.4% | 79.8% | 23.7% |
> | **Self-RedTeam (Llama3.1-8B-IT)** | 50.8% | 54.8% | 78.6% | 13.5% |

---

### Author Response · Authors · 2025-12-02
**Rebuttal Summary**

We would like to thank the reviewers for their hard work in reviewing this paper. We understand that the recent ICLR identity leak may have impacted the reviewers' willingness to participate in the rebuttal process. We fully support the integrity of the double-blind review system and the importance of protecting their anonymity. For your convenience, we have summarized our responses to the reviewers' comments below:

---

### **The reviewers collectively praised the paper's execution and conceptual novelty.**

- Reviewer h9EC highlighted that the method is ***"practical and efficient"*** and that the paper is ***"very well written and polished"***.
- Reviewer 4Pif described the framework as a ***"conceptually appealing formulation"*** and noted that the Hidden CoT mechanism is ***"an elegant addition"***.
- Reviewer mcqS commended the ***"strong empirical results"*** and the ***"extensive ablations"***  that demonstrated the effectiveness of the proposed components.

### **We addressed the shared concerns regarding evaluation depth, reward model dependence, and model architecture as follows**:

> Reviewer h9EC (6) and Reviewer 4Pif (6) expressed concern that our evaluations were primarily against static adversarial prompts and asked for stronger, dynamic non-static attacks.

We agreed and conducted extensive new dynamic evaluations. We highlighted the existing results on X-Teaming and added results for **PAIR** and **AutoDAN**, which all of these three are dynamic multi-turn attack methods, in the revised Appendix C.4. These results demonstrate that our method achieves **significant robustness gains** (e.g., up to 50.7% ASR reduction on PAIR and AutoDAN) and generalizes well to adaptive adversaries. We only included the X-Teaming results originally because we believe that X-Teaming being a much more recent work is closer for being a SOTA attack method, and dynamic evaluation methods have enormous computational demands.

> Reviewer h9EC (6) and Reviewer 4Pif (6) raised concerns about potential reward hacking and the method's sensitivity to the specific reward model used (WildGuard).

To prove our method is not overfitting to a specific judge, we ran new experiments replacing WildGuard with **Qwen3Guard** (a stronger, more recent model). As shown in the new tables in Appendix D.6, the results remained consistent if not better, confirming that Self-RedTeam is robust to different reward models and does not rely on exploiting specific blind spots.

> Reviewer mcqS (4) and Reviewer h9EC (6) questioned the rationale behind using a single shared model for both attacker and defender roles, rather than separate models.

We clarified in Appendix E that this design is intentional for two reasons: 1) Computational Accessibility, as running two online models simultaneously doubles the computational overheads and 2) Autonomous Self-Improvement, allowing the model to surface and correct its own blind spots rather than overfitting to an external adversary's distribution.

> Reviewer h9EC (6) raised questions regarding the emergent attacker refusal failure case presented in Appendix G.2.

We clarified that **this behavior is a model-specific artifact intrinsic to Llama-3.1's base safety alignment** rather than a methodological flaw. We conducted a quantitative analysis on 3,000–5,000 revised attacks and found that all Qwen2.5-based Self-RedTeam checkpoints exhibited negligible refusal rates (<0.11%), whereas Llama-3.1-8B-based Self-RedTeam checkpoints retained a 25.27% refusal rate.

> Reviewer 4Pif (6) worried that the evaluation benchmarks (WildGuard/WildJailbreak) might overlap with training data, potentially inflating results.

We clarified that there is strictly **no data leakage**. Our training uses "vanilla" harmful seeds, whereas the evaluation is performed solely on "adversarial" prompts from the test split that the model never sees during training. Additionally, "vanilla" seeds are usually one short sentences, where as "adversarial" prompts are paragraphs consisted of multiple sentences.

> Reviewer h9EC (6) and Reviewer 4Pif (6) felt the Nash Equilibrium (NE) theoretical discussion seemed "a bit ad-hoc" regarding "convergence... in practice".

We expanded our discussion to clarify that the NE analysis serves as a theoretical "North Star" for guiding the training objective rather than a guaranteed end-state as achieving it in pracitice is very difficult in non-convex deep learning. But on the bright side as we pointed to Figure 3, which shows empirical stability where the defender's win rate approaches a fixed point near 100%, mirroring NE properties.

> Reviewer mcqS (4) noted confusion regarding the notation of shared parameters and undefined KL terms.

We acknowledged the ambiguity and updated Section 3 and Eq. 1 to explicitly distinguish between the reward model (parameterized by $\phi$) and the policy model (parameterized by $\theta$), and we explicitly defined the token-wise reverse KL divergence term.

---

### Meta-Review · Area_Chair_cnh7 · 2026-01-06

**Summary:**

This paper proposes Self-RedTeam, an online self-play RL setup where a single model alternates between attacker and defender roles (with hidden CoT), aiming to move from reactive “patching” toward proactive co-evolution for safer LMs. The paper also motivates the approach via a two-player zero-sum framing, with a “Nash equilibrium as north star” style safety argument, and reports broad empirical gains across model families and safety benchmarks.

**Reviewer Concerns:**

Across reviews, the main positives were: (i) a clean and practical formulation, (ii) strong empirical coverage/ablations, and (iii) good writing/polish. Reviewer h9EC in particular was leaning weak accept and highlighted practicality/efficiency and overall execution quality, while flagging concerns around evaluation strength (dynamic/adaptive attacks), reward-model dependence, and the single-model attacker/defender design. Reviewer 4Pif similarly rated the work slightly above threshold and raised overlapping concerns about reward-model reliance, evaluation realism, and potential benchmark overlap.

**Reviewer Scores:**

Reviewer h9EC: likely no change
Reviewer 4Pif: likely no change
Reviewer mcqS: likely no change

---

### Decision · Program_Chairs · 2026-01-26

Reject